

# One-step multiplex real-time RT-PCR assay for detecting and genotyping wild-type group A rotavirus strains and vaccine strains (Rotarix® and RotaTeq®) in stool samples

Rashi Gautam, Slavica Mijatovic-Rustempasic, Mathew D. Esona, Ka Ian Tam, Osbourne Quaye and Michael D. Bowen

Division of Viral Diseases, Gastroenteritis and Respiratory Viruses Laboratory Branch, Centers for Disease Control and Prevention, Atlanta, Georgia, United States of America

Corresponding author
Rashi Gautam, IJS0@cdc.gov

## ABSTRACT

**Background.** Group A rotavirus (RVA) infection is the major cause of acute gastroenteritis (AGE) in young children worldwide. Introduction of two live-attenuated rotavirus vaccines, RotaTeq® and Rotarix®, has dramatically reduced RVA associated AGE and mortality in developed as well as in many developing countries. High-throughput methods are needed to genotype rotavirus wild-type strains and to identify vaccine strains in stool samples. Quantitative RT-PCR assays (qRT-PCR) offer several advantages including increased sensitivity, higher throughput, and faster turnaround time. **Methods.** In this study, a one-step multiplex qRT-PCR assay was developed to detect and genotype wild-type strains and vaccine (Rotarix® and RotaTeq®) rotavirus strains along with an internal processing control (Xeno or MS2 RNA). Real-time RT-PCR assays were designed for VP7 (G1, G2, G3, G4, G9, G12) and VP4 (P[4], P[6] and P[8]) genotypes. The multiplex qRT-PCR assay also included previously published NSP3 qRT-PCR for rotavirus detection and Rotarix® NSP2 and RotaTeq® VP6 qRT-PCRs for detection of Rotarix® and RotaTeq® vaccine strains respectively. The multiplex qRT-PCR assay was validated using 853 sequence confirmed stool samples and 24 lab cultured strains of different rotavirus genotypes. By using thermostable *rTth* polymerase enzyme, dsRNA denaturation, reverse transcription (RT) and amplification (PCR) steps were performed in single tube by uninterrupted thermocycling profile to reduce chances of sample cross contamination and for rapid generation of results. For quantification, standard curves were generated using dsRNA transcripts derived from RVA gene segments. **Results.** The VP7 qRT-PCRs exhibited 98.8–100% sensitivity, 99.7–100% specificity, 85–95% efficiency and a limit of detection of 4–60 copies per singleplex reaction. The VP7 qRT-PCRs exhibited 81–92% efficiency and limit of detection of 150–600 copies in multiplex reactions. The VP4 qRT-PCRs exhibited 98.8–100% sensitivity, 100% specificity, 86–89% efficiency and a limit of detection of 12–400 copies per singleplex reactions. The VP4 qRT-PCRs exhibited 82–90% efficiency and limit of detection of 120–4000 copies in multiplex reaction. **Discussion.** The one-step multiplex qRT-PCR assay will facilitate high-throughput rotavirus genotype characterization for monitoring circulating rotavirus wild-type

strains causing rotavirus infections, determining the frequency of Rotarix® and RotaTeq® vaccine strains and vaccine-derived reassortants associated with AGE, and help to identify novel rotavirus strains derived by reassortment between vaccine and wild-type strains.

## INTRODUCTION

Group A rotavirus (RVA) infection is the major etiologic agent of acute gastroenteritis (AGE) in children aged <5 y worldwide and is associated with an estimated 453,000 deaths annually, predominantly in developing countries (*Tate et al., 2012*). RVA belongs to the *Reoviridae* family and its genome consists of 11 double-stranded RNA (dsRNA) segments which encode six structural proteins (VP1-VP4, VP6 and VP7) and five or six non-structural proteins (NSP1-NSP5/NSP6) (*Estes & Kapikian, 2007*). Rotaviruses are classified based on the serological characteristics or sequence diversity of two outer capsid proteins, VP7 (glycosylated, G-type) and VP4 (protease sensitive, P-type) (*Iturriza-Gomara, Kang & Gray, 2004*). Till date, at least 27 G- and 37 P-genotypes have been recognized and approximately 73 G/P genotype constellations of RVAs infecting humans have been reported (*Matthijnssens et al., 2011*; *Trojnar et al., 2013*). Of all the possible combinations, 6 genotypes (G1P[8], G2P[4], G3P[8], G4P[8], G9P[8], and G12P[8]) contribute to an estimated 80–90% of the global RVA disease burden (*Matthijnssens et al., 2009*; *Banyai et al., 2012*; *Patel et al., 2011*; *Iturriza-Gomara et al., 2011*). Two live attenuated oral vaccines, Rotarix® (GlaxoSmithKline Inc., Rixensart, Belgium) and RotaTeq® (Merck, Blue Bell, PA, USA) provide protection against severe diarrhea caused by the major RVA serotypes in circulation and has dramatically reduced childhood AGE in developed as well as in many developing countries (*Patel et al., 2011*). However, Rotarix® and RotaTeq® are live vaccines that can replicate in vaccinees and are shed in faeces post vaccination (*Anderson, 2008*; *Yen et al., 2011*). In addition RotaTeq® component strains can reassort with one another to produce reassortant strains causing gastroenteritis (*Bowen & Payne, 2012*). Reassortant strains derived from vaccine strains Rotarix® (*Rose et al., 2013*) and RotaTeq® (*Patel et al., 2010*; *Werther et al., 2009*) have been associated with AGE in vaccinated (*Bowen & Payne, 2012*; *Boom et al., 2012*; *Hemming & Vesikari, 2012*; *Donato et al., 2012*; *Bucardo et al., 2012*) and unvaccinated children (*Payne et al., 2010*; *Rivera et al., 2011*). Rotavirus genotype characterization is important to monitor circulating rotavirus wild-type strains causing rotavirus infections, to detect the frequency of Rotarix® and RotaTeq® vaccine strain components and vaccine derived reassortants associated with AGE, and to identify novel rotavirus strains derived by reassortment and interspecies transmission of rotavirus strains from animals to humans (*Fischer & Gentsch, 2004*).

Molecular techniques are considered the 'gold standard' for genotyping of rotavirus strains. Standard methods used in the characterization (G and P genotypes) of rotavirus strains include enzyme linked immunosorbent assay (ELISA) serotyping (*Gomara, Green & Gray, 2000*), microarray hybridization (*Chizhikov et al., 2002*), restriction fragment length polymorphism (RFLP) (*Iturriza Gomara et al., 2002*), one-step or two-step conventional reverse transcription-polymerase chain reaction (RT-PCR) followed by gel based genotyping of PCR amplicons (*Gouvea et al., 1990*; *Gentsch et al., 1992*; *Das et al., 1994*) and nucleotide sequencing (*DiStefano et al., 2005*). Conventional methods to genotype rotavirus strains are costly, labor intensive with low throughput and low limit of detection resulting in non-typeable or incompletely typed strains (*Fischer & Gentsch, 2004*). Real time RT-PCR assays (qRT-PCRs) offer several advantages over traditional RT-PCR, including increased sensitivity, higher throughput, faster turnaround time, quantification of viral RNA and less risk of sample cross-contamination due to elimination of post-amplification product manipulation (*Mijatovic-Rustempasic et al., 2013*). Several singleplex qRT-PCR assays have been developed for detection of RVA targeting VP2 (*Gutierrez-Aguirre et al., 2008*), VP4 (*Min et al., 2006*; *Kottaridi et al., 2012*), VP6 (*Logan, O'Leary & O'Sullivan, 2006*; *Kang et al., 2004*; *Nordgren et al., 2010*), VP7 (*Kottaridi et al., 2012*; *Plante et al., 2011*), NSP3 (*Freeman et al., 2008*; *Jothikumar, Kang & Hill, 2009*; *Mijatovic-Rustempasic et al., 2013*), NSP4 (*Adlhoch et al., 2011*) genes, Rotarix® and RotaTeq® vaccine strains (*Gautam et al., 2013*) but multiplex qRT-PCR assays are not widely reported for detection and genotyping of rotavirus strains. Multiplex RT-PCR Luminex assay (*Liu et al., 2011*) and TaqMan array card (*Liu et al., 2013*) assays have been developed for simultaneous detection and quantitation of enteropathogens in stool samples including rotavirus. A multiplex RT-PCR reverse hybridization strip assay has been developed to genotype VP7 and VP4 strains in stool samples but, this method involves numerous steps, is time consuming and requires visual interpretation of results (*van Doorn et al., 2009*). Multiplex qRT-PCR assay to genotype only 4 G-types and 2 P-types have been developed, but there are little data showing the performance of these newly developed assays (efficiency, limit of detection, sensitivity, specificity) or confirmation of multiplex qRT-PCR genotyping results with a gold standard method (*Podkolzin et al., 2009*). Two multiplex qRT-PCR assays to genotype VP7 (G1, G2, G3, G4 and G9 strains) and VP4 (P[4] and P[8]) RVA strains (*Kottaridi et al., 2012*) and VP7 (G1, G2, G3, G4, G8, G9, G10 and G12 strains) and VP4 (P[4], P[6], P[8], P[10] and P[11]) (*Liu et al., 2014*) have been reported in the literature but the newly developed assays were tested on limited number of clinical samples (~90) and also involved two or three step qRT-PCR approach involving separate denaturation, reverse transcription and amplification steps (*Kottaridi et al., 2012*; *Liu et al., 2014*).

A one step multiplex qRT-PCR assay to genotype VP7 (G1, G2, G3, G4, G9 and G12) and VP4 (P[4], P[6], P[8]) strains has not been reported in the literature so far. In this study, we have developed and validated a one-step multiplex qRT-PCR assay (4 wells, 13 component assays) to detect and genotype wild-type and vaccine (Rotarix® and RotaTeq®) rotavirus strains along with an internal processing control. As RT-PCR assays
for rotavirus detection and genotyping are prone to false-negative results due to inhibitory substances in faeces, we have incorporated Xeno-armored RNA or MS2 as internal processing controls (IPC) to monitor extraction efficiency and to detect inhibition. We have developed the multiplex qRT-PCR genotyping assay using a thermostable *rTth* polymerase enzyme in order to perform denaturation of dsRNA followed by qRT-PCR in the same tube (*Mijatovic-Rustempasic et al., 2013*; *Gautam et al., 2013*). In addition, we have generated artificial dsRNA positive control transcripts for VP7 and VP4 genes and quantitated each VP7 (G1, G2, G3, G4, G9 and G12) and VP4 (P[4], P[6], and P[8]) assay using their respective transcripts. The multiplex qRT-PCR assay was validated on large number of sequence confirmed stool samples (n = 853) and lab cultured strains (n = 24) of different genotypes. The one-step multiplex qRT-PCR assay developed in this study can simultaneously detect and genotype wild-type rotavirus strains and vaccine strains in stool samples.

## MATERIALS AND METHODS

### Primer and probe design

The consensus sequences obtained from multiple alignments (using MEGA version 5 software- http://www.megasoftware.net/, GeneDoc. and Multalign (*Corpet, 1988*)) of VP7 (G1, G2, G3, G4, G9, G12) and VP4 (P[4], P[6], P[8]) genes were used to design TaqMan assays for each gene which targeted gene regions and nucleotide substitutions specific to each genotype. Based on the multiple alignments of VP7 and VP4 gene sequences, universal G-forward and P-forward primers were designed at the 5′ end of VP7 and VP4 genes, respectively. Thus, same VP7 forward primer was used for all VP7 (G1, G2, G3, G4, G9 and G12) qRT-PCRs and same VP4 forward primer was used for all VP4 (P[4], P[6] and P[8]) qRT-PCRs (Table 1). Probes and reverse primers were designed targeting the nucleotides specific for VP7 (G1, G2, G3, G4, G9 and G12) and VP4 (P[4], P[6], and P[8]) genotypes. Multiple probe and primer sets were designed manually and degenerate bases were introduced into the primer or probe sequences to account for nucleotide variation among same genotype strains observed in sequence alignments of above-mentioned genes. The primer and probe sequences were checked for specificity using NCBI-Nucleotide blast (nblast) (http://blast.ncbi.nlm.nih.gov/Blast.cgi) and were checked for self-annealing sites, hairpin loop formation and 3′ complementarity using the IDT oligonucleotide calculator (http://www.idtdna.com/analyzer/Applications/OligoAnalyzer/). The melting temperatures ($T_m$) of primers and probes were increased to 57–60 °C and 63–70 °C, respectively, using C-5 propynyl-dC (pDC) or AP-dC (G-clamp) substitutions as necessary (Glen Research, CA, USA). Primers and probe sequences for detection of rotavirus (NSP3 assay) and rotavirus vaccine strains (Rotarix® NSP2 and RotaTeq® VP6 assays) were published previously (*Mijatovic-Rustempasic et al., 2013*; *Gautam et al., 2013*). Primer and probe sequences for internal process control (Xeno) were proprietary and were obtained from Life Technologies Corp., Grand Island, NY, USA. Primer and probe sequences for internal process control (MS2) were obtained from published literature (*Rolfe et al., 2007*). All primers and probes were synthesized by the

**Table 1  Primers and probes designed for multiplex qRT-PCR rotavirus genotyping assay.**

| | Target real-time assays | Nucleotide sequence (5′-3′)[a] | Fluorophore-quencher (5′-3′) | Conc./qRT-PCR reaction (nM) | Nucleotide position (bp) | Tm (°C) | Amplicon length (bp) |
|---|---|---|---|---|---|---|---|
| **Well 1** | RotaTeq® VP6 | FP[b]-GCGGCGTTATTTCCAAATGCACAG | | 200 | *Gautam et al., 2013* | | |
| | | RP[c]-CGTCGGCAAGCACTGATTCACAAA | | 200 | | | |
| | | Probe[e]-ATCACGCAA″C″AGTAGGACT″C″ACGCTT | HEX-BHQ1 | 100 | | | |
| | Xeno-IPC | FP[b]-Proprietary | | 400 | Proprietary | | |
| | | RP[c]-Proprietary | | 400 | | | |
| | | Probe[e]-Proprietary | TR-BHQ2 | 100 | | | |
| | MS2-IPC | FP[b]-TGGCACTACCCCTCTCCGTATTCACG | | 400 | 289–314 | 63 | 99 |
| | | RP[c]-GTACGGGCGACCCCACGATGAC | | 400 | 387–366 | 64 | |
| | | Probe[e]-CACATCGATAGATCAAGGTGCCTACAAGC | TR-BHQ2 | 200 | 330–358 | 60 | |
| | Rotarix® NSP2 | FP[b]-GAACTTCCTTGAATATAAGATCACACTGA | | 400 | *Gautam et al., 2013* | | |
| | | RP[c]-TTGAAGACGTAAATGCATACCAATTC | | 400 | | | |
| | | Probe[e]-TCCAATAGATTGAAGT{C}AGTAA″C″GTTTCCA | Cy5-BHQ3 | 200 | | | |
| | G12 | G-consensus-FP[b]-TAG{C}TCYTTTTRATGTATGGTAT | | 200 | 37–59 | 57 | 309 |
| | | RP[c]-CGTCCARTCRGGRTCAGTTATTTCAGTC | | 200 | 345–318 | 60 | |
| | | Probe[d]-ARTTTTGAG{C}TYYAATAAATGGCAGYAYG | FAM-BHQ1 | 100 | 205–177 | 65 | |
| **Well 2** | G9 | G-consensus-FP[b]-TAG{C}TCYTTTTRATGTATGGTAT | | 600 | 37–59 | 57 | 318 |
| | | RP[c]-CAGAGTATYTT″C″CATTCHGTATCTCC | | 600 | 354–328 | 58 | |
| | | Probe[e]-CCACARTT{C}TAA{C}CTTTYTGATATCA | HEX-BHQ1 | 200 | 68–93 | 68 | |
| | NSP3 | FP[b]-ACCATCTWCACRTRACCCTC | | 400 | *Mijatovic et al., 2013* | | |
| | | RP[c]-GGTCACATAACGCCCCTATA | | 400 | | | |
| | | Probe[e,f]-ATGAGCACAATAGT″T″AAAAGCTAACACTGTCAA | TR-BHQ2 | 100 | | | |
| | G4 | G-consensus-FP[b]-TAG{C}TCYTTTTRATGTATGGTAT | | 600 | 37–59 | 57 | 319 |
| | | RP[c]-ATAGWGTAT{C}TTTCCATTCAKTGTC | | 600 | 355–331 | 59 | |
| | | Probe[e]-TGTAGTATTRT{C}AKTATTAK{C}GAATGCRC | FAM-BHQ1 | 200 | 171–199 | 70 | |
| **Well 3** | G1 | G-consensus-FP[b]-TAG{C}TCYTTTTRATGTATGGTAT | | 400 | 37–59 | 57 | 305 |
| | | RP[c]-CAKTCACCATCATTRATTTGAGTACTTGCT | | 400 | 341–322 | 57 | |
| | | Probe[d]-TCCATTGAT{C}CTGTTATTGGTAAGTTAAG | HEX-BHQ1 | 100 | 239–210 | 63 | |
| | P4 | P-consensus-FP[b]-GG″C″TATAAAATGG″C″TTCGCT | | 400 | 1–20 | 59 | 459 |
| | | RP[c]-ATYACYMTGACTACTAC″C″TTTAAA″C″ATTTCG | | 400 | 459–429 | 60 | |
| | | Probe[d,f]-ATGTGGT″T″CRACWGCGATAACTG″C″TGTC ″C″AAAA | TR-BHQ2 | 100 | 342–310 | 69 | |
| | G3 | G-consensus-FP[b]-TAG{C}TCYTTTTRATGTATGGTAT | | 400 | 37–59 | 57 | 459 |
| | | RP[c]-TTGCAGTGTAG″C″GTCRTAYTTC | | 400 | 495–474 | 58 | |
| | | G3-P[e]-TGTATTAYC{C}AACTGAAG″C″WGCAACAG | Cy5-BHQ3 | 100 | 296–322 | 69 | |
| **Well 4** | G2 | G-consensus-FP[b]-TAG{C}TCYTTTTRATGTATGGTAT | | 200 | 37–59 | 57 | 352 |
| | | RP[c]CARTYGGCCAT{C}CTTTAGTTA | | 200 | 388–378 | 60 | |
| | | Probe[e]-CCAATAACGGGRT{C}ACTAGACGCTGT | TR-BHQ2 | 100 | 220–246 | 69 | |

(Continued)

| Target real-time assays | Nucleotide sequence (5′-3′)[a] | Fluorophore-quencher (5′-3′) | Conc./qRT-PCR reaction (nM) | Nucleotide position (bp) | Tm (°C) | Amplicon length (bp) |
|---|---|---|---|---|---|---|
| P8 | P-consensus-FP[b]-GG″C″TATAAAATGG″C″TTCGCT | | 400 | 1–20 | 59 | 424 |
| | RP[c]-ACTTCCAYTTAT{C}TGAATCRTTWCT | | 400 | 424–400 | 60 | |
| | Probe[e]-A{C}TGYAGTMGTTG″C″TRTTGAACCRCA | Cy5-BHQ3 | 200 | 316–341 | 70 | |
| P6 | P-consensus-FP[b]-GG″C″TATAAAATGG″C″TTCGCT | | 400 | 1–20 | 59 | 297 |
| | RP[c]-AAGCAAYCCAAAYAT{C}AGTTTTATTG | | 400 | 297–322 | 68 | |
| | Probe[e]TGAATC{C}AACTAATCAA{C}AAGTTG | FAM-BHQ1 | 200 | 260–283 | 68 | |

**Notes:**
[a] {C}, AP-dC (G-clamp); "C," C-5 propynyl-dC (pDC); "Y"-C or T; "R"-A or G; "M"-C or A; "K"-T or G; "W"-T or A.
[b] Forward primer.
[c] Reverse primer.
[d] Probe in reverse compliment orientation.
[e] Probe in forward orientation.
[f] Probe with internal quencher at "T" nucleotide. IPC-Internal Positive Control.
HEX- 5′-Hexachloro-Fluorescein Phosphoramidite- (Glen Research, Catalog Number- 10-5902-95).
CalRed 610 or Texas Red (TR) - (BioSearch Technologies, Catalog Number-BNS-5082-50).
Quasar 670 or Cy5- (BioSearch Technologies, Catalog Number-FC-1065-100).
FAM-5′ carboxyfluorescein- (Glen Research, Catalog number- 10-5901-95).
BHQ1- Black Hole Quencher-1-(BioSearch Technologies, Catalog Number-3-BHQ1-1).
BHQ2-Black Hole Quencher-2-(BioSearch Technologies, Catalog Number-CG5-5042G-2).
BHQ3-Black Hole Quencher-3-(BioSearch Technologies, Catalog Number-CG5-5043G-2).

Biotechnology Core Facility at the Centers Disease Control and Prevention (CDC), Atlanta, GA or BioSearch Technologies, Novato, CA, USA.

## Internal process control and RNA extractions

Ten percent suspensions (weight/volume for solid stools and volume/volume for liquid stools) of each sample were prepared in phosphate-buffered saline (PBS), from stool samples and reference strains. To introduce an internal process control into the multiplex qRT-PCR assays, 2 µL of $10^8$ copies/uL of Xeno RNA (Life Technologies Corp., Grand Island, NY, USA) or 2 µL of $10^9$ units/uL of MS2 bacteriophage RNA (ZeptoMetrix, Buffalo, NY, USA) were spiked into a 98 µL volume of 10% stool suspension prepared in PBS. The assay can use either process control, Xeno RNA (proprietary) or MS2 bacteriophage, and can be detected using Xeno (proprietary) or MS2 (*Liu et al., 2011*) qRT-PCR assays respectively. The samples with spiked Xeno/MS2 were extracted using the MagMax 96 Viral RNA Isolation kit (Applied Biosystems, Inc., Foster City, CA, USA) on an automated KingFisher extraction system (Thermo Scientific, Waltham MA) or the MagNA Pure Compact RNA extraction kit on MagNA Pure Compact instrument (Roche Applied Science, Indianapolis, IN, USA) according to manufacturer's instructions. Stool samples (n = 853) containing either wild-type RVA, vaccine-strains Rotarix® (n = 39), RotaTeq® (n = 83), or RVA-negative (n = 20) were obtained from routine domestic and international RVA surveillance conducted by CDC. RVA laboratory cultured strains (n = 24) representing various VP7 (G) and VP4 (P) genotypes were also used to screen, optimize and validate the multiplex qRT-PCR assay. The lab cultured strains used were: Wa (G1P[8]), DS-1 (G2P[4]), P(G3P[8]), ST3 (G4P[6]), 116E (G9P[11]), Wi61(G9P[8]),

US1205 (G9P[6]), L26 (G12P[4]), 1076 (G2P[6]), F45 (G9P[8]), AU-1 (G3P[9]), I321 (G10P[11]), CC117 (G9P[8]), 69M (G8P[10]), NCDV (G6P[1]), WC3 (G6P[5]), B223 (G10P[11]), L338 (G13P[18]), OSU G5P[7], EDIM (G16P[16]), RRV (G3P[3]), SA11 (G3P[2]), CC425 (G3P[9]) and RO1845 (G3P[3]). All clinical stool samples were deidentified and could not be traced back to patient or hospital case identifiers. Non-RVA gastroenteritis virus strains, including norovirus, sapovirus, astrovirus, and adenovirus were used to validate the multiplex qRT-PCR assay. All extracted RNAs were stored at −80 °C until analyzed.

## Conventional RT-PCR and sequencing

To confirm the VP7 and VP4 genotype of each sample and to confirm Rotarix® and RotaTeq® vaccine strains in samples, RT-PCR of VP7, VP4 (*Das et al., 1994*; *Gentsch et al., 1992*; *Gomara et al., 2001*; *Gouvea et al., 1990*), NSP2 (*Gomara, Green & Gray, 2000*; *Matthijnssens et al., 2006*), VP6 (*Gomara, Green & Gray, 2000*), genes were performed and the resulting amplicons were sequenced by using an ABI 3130 xl sequencer. The consensus sequence of each gene was queried to the GenBank sequence database by using nucleotide BLAST (http://blast.ncbi.nlm.nih.gov/Blast.cgi) or RotaC online classification tool (http://rotac.regatools.be/) (*Maes et al., 2009*) for genotype characterization.

## Screening and optimization of singleplex qRT-PCR assays

Multiple probes and primer sets were designed for each VP7 (G1, G2, G3, G4, G9, G12) and VP4 (P[4], P[6], P[8]) genotypes. All qRT-PCR assays designed were screened using lab cultured positive strains and a small panel of clinical stool samples of various genotypes including G1P[8], G2P[4], G3P[8], G4P[6], G9P[8], G12P[8]. For each real-time assay, the probe and primer set with the best sensitivity, specificity, efficiency and limit of detection as determined using artificially constructed dsRNA transcripts was selected for optimization. For assays having more than one specific probe and primer pair, the probe and primer set showing a sigmoidal amplification curve with the lowest quantification cycle (Cq) value was selected for optimization. Selected probe and primer sets were optimized by performing each assay at four different probe and primer concentrations (100:200 nM, 100:400 nM and 200:400 nM and 200:600 nM). The probe and primer concentration for each assay showing amplification of its respective template at the lowest Cq value with low background amplification was selected for subsequent experiments (Table 1).

All qRT-PCR experiments were performed in 96-well ABI Fast plates using the GeneAmp EZ rTth RNA PCR kit (Applied Biosystems, Inc., Foster City, CA, USA). Single well denaturation, reverse transcription and amplification were performed using a 7500 Fast Real-Time PCR System in fast mode (Applied Biosystems, Inc., Foster City, CA, USA).

## Double stranded RNA (dsRNA) positive control transcripts

Primers with T7 promoter sequences at the 5′ end of both the forward and reverse primers were designed to generate full length amplicons for VP7 (G1, G2, G3, G4, G9 and G12)

and VP4 (P[4], P[6] and P[8]) genes (Table S1). The transcription template for VP7 genotype strains were prepared by using T7 tailed primer pairs for the VP7 gene and their respective RNA as template (Wa-G1, DS-1-G2, P-G3, ST3-G4, WI61-G9 and L26-G12). The transcription template for VP4 genotype strains were prepared by using T7 tailed primer pairs for the VP4 gene and their respective RNA as template (L26-P[4], ST3-P[6], and Wa-P[8]). After heat-denaturing the RNAs for 5 min at 95 °C, the QIAGEN One Step RT-PCR kit (QIAGEN, Inc., Valencia CA, USA) was used to perform RT-PCR as per manufacturer's instructions with following cycling conditions: 30 min at 42 °C; 15 min at 95 °C; 30 cycles of 30 sec at 94 °C, 30 sec at 65 °C and 45 sec at 72 °C; 7 min at 72 °C; and 4 °C hold. The resulting amplicon was analyzed on a 1% agarose gel and the specific band was purified using QIA quick Gel Extraction kit (QIAGEN, Venlo, Netherlands). The Megascript® RNAi kit (Ambion®, Grand Island, NY, USA) was used to generate dsRNA transcript according to manufacturer's instructions. Concentrations of the transcripts were measured at 260 nm using a NanoDrop ND-1000 spectrophotometer (Thermo Scientific, Wilmington, DE, USA). The efficiency and limit of detection of VP7 and VP4 qRT-PCRs were determined by analyzing VP7 (G1, G2, G3, G4, G9 and G12) and VP4 (P[4], P[6] and P[8]) qRT-PCRs using ten-fold dilution series of their respective transcripts. Ten-fold dilution series of all the transcripts ($10^{-2}$ to $10^{-12}$) were prepared in DEPC-treated water containing 100 ng/µL yeast carrier RNA (Ambion, Grand Island, NY, USA). A standard curve was generated by plotting the log copy number against Cq value and was fitted with a regression line. The slope for calculation of efficiency was obtained from the regression line. Copy number of each assay was calculated by using the formula:

$$\text{Copy number (molecules}/\mu L) = [\text{concentration (ng}/\mu L) \times 6.022 \times 10^{23} \text{ (molecules/mol)}] / [\text{length of amplicon} \times 650 \text{ (g/mol)} \times 10^9 \text{ (ng/g)}].$$

## Multiplex qRT-PCR assay

Each sample was tested by multiplex qRT-PCR assay in four separate reactions. The master mix for each well contained 5 µL of 5X EZ buffer, 2.4 µM dNTPs, 2.5 mM Mn(OC)$_2$, 2.5 U/µL *rTth* polymerase (wells 1, 2 and 3) or 5.0 U/µL *rTth* polymerase (Well 4), forward primer, reverse primer and probe at final concentrations specific for each component assay (Table 1) and 2 µL template (RNA or nuclease free water) in a 25 µL final reaction volume. All the samples and negative controls were tested in duplicate. The cycling conditions consisted of denaturation of dsRNA for 5 min at 95 °C, reverse transcription for 30 min at 50 °C, 1 min at 95 °C and 45 amplification cycles consisting of 15 sec at 95 °C and for 1 min at 60 °C.

Selection and optimization of the probe was performed first using FAM reporter dye. Subsequently, the selected probes were synthesized using different reporter dyes such as HEX, Texas Red (Cal Red 610) and Cy5 (Quasar 670). For published NSP3, Rotarix® NSP2 and RotaTeq® VP6 assays, the same probe sequences were used with different reporter dyes. The probes with all 4 reporter dyes/probe (FAM, HEX, Texas Red and Cy5) were again tested on lab cultured positive strains and transcript ten-fold dilutions for efficiency and limit of detection. For each genotype, the probe with reporter dye showing a

sigmoidal amplification curve with the lowest Cq value was selected for the multiplex qRT-PCR assay. After selection of reporter dye and designation of a well number to each qRT-PCR with least background and minimum cross reactivity, all the 13 qRT-PCRs were validated in multiplex format (4 wells) using sequence confirmed clinical samples (~100 samples/genotype), 24 reference strains and 2 vaccine strains, Rotarix® (39 samples) and RotaTeq® (83 samples).

**Assay Performance calculations.** The sensitivity, specificity, positive predictive value (PPV) and negative predictive value (NPV) of each qRT-PCR assay were calculated using standard procedures (*Dawson & Trapp, 1994*).

## Ethics statement

For domestic surveillance samples, institutional review board approvals were obtained from the CDC and from individual study sites or were determined to be exempt from CDC institutional review board approval because case identification information was not collected. All clinical samples tested in this study, from both domestic and international surveillance, were de-identified so they could not be linked to cases.

## RESULTS

### Multiplex qRT-PCR assay

For developing the qRT-PCR assay to genotype rotavirus VP7 and VP4 genes, singleplex qRT-PCRs were first designed, optimized and validated on lab cultured rotavirus strains of various genotypes. Validated singleplex qRT-PCRs for VP7 (n = 6), VP4 (n = 3) were then combined in the four well multiplex format which also included previously published NSP3 qRT-PCR for rotavirus detection, RotaTeq® VP6 qRT-PCR to detect RotaTeq® vaccine strains, Rotarix® NSP2 qRT-PCR to detect Rotarix® vaccine strain and a Xeno RNA or MS2 qRT-PCR for an internal process control. Multiplex qRT-PCR assay (4 wells/13 qRT-PCRs) were tested on 853 sequence confirmed clinical samples.

### Well 1 multiplex qRT-PCRs

Multiplex well 1 consisted of RotaTeq® VP6 HEX, Xeno or MS2-TR, Rotarix® Cy5 and G12-FAM qRT-PCR. When tested on Rotarix® and RotaTeq® vaccine strains and lab cultured positive control strain L26 for G12 genotype, multiplex well 1 qRT-PCRs showed amplification with RotaTeq® vaccine strain, samples spiked with Xeno RNA at Cq 29–32, Rotarix® vaccine strain and L26 RNA (Fig. 1, Well 1A). Positive control samples spiked with MS2 RNA showed amplification with MS2-TR at Cq 27–30 (Fig. 1, Well 1B).

#### *RotaTeq® VP6 HEX*

The RotaTeq® VP6 HEX qRT-PCR detected RotaTeq® vaccine strains in all sequence-confirmed RotaTeq® positive samples (n = 83) tested (Table 2-Well 1A, Fig. 2A). The RotaTeq® VP6 HEX qRT-PCR showed no amplification with samples of other genotypes (n = 750), or with RVA negative samples (n = 20) (Table 2-Well 1A). When tested using laboratory reference strains (n = 24) and 2 vaccine strains (Rotarix® and RotaTeq®), the RotaTeq® VP6 HEX qRT-PCR showed amplification with RotaTeq® vaccine strain, bovine strains (B223, WC3, NCDV and RRV) and human-animal reassortant strains

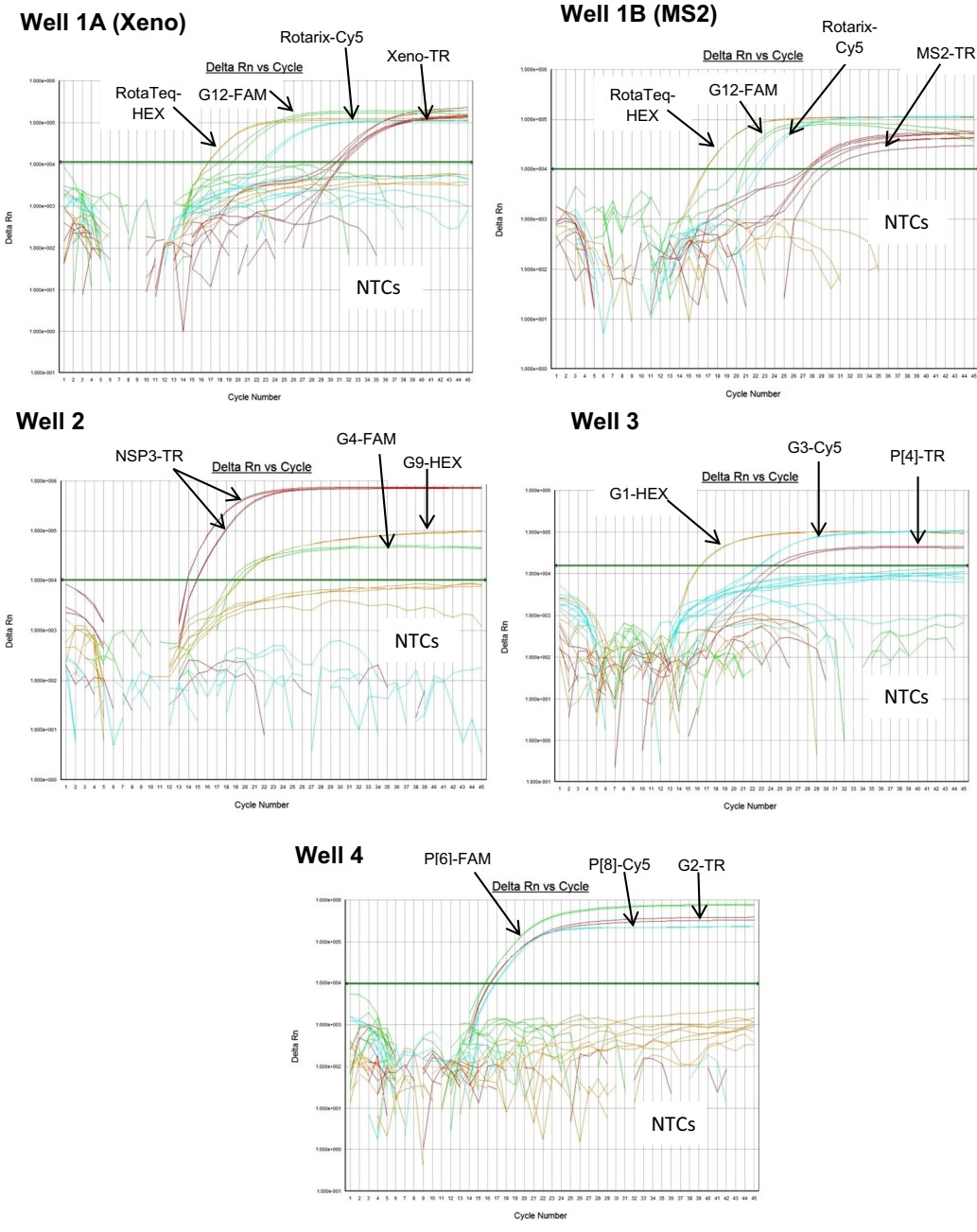

**Figure 1 Performance of 4 well multiplex qRT-PCR assay on lab cultured positive controls.** Well 1A (Xeno) - Amplification plots of Xeno-TR, RotaTeq®-HEX, Rotarix®-Cy5, and G12-FAM qRT-PCR on samples spiked with Xeno, RotaTeq® vaccine strain, Rotarix® vaccine strain and strain L26. Well 1B (MS2) - Amplification plots of MS2-TR, RotaTeq®-HEX, Rotarix®-Cy5, and G12-FAM assays on samples spiked with MS2, RotaTeq® vaccine strain, Rotarix® vaccine strain and strain L26. Well 2-Amplification plots of NSP3-TR, G9-HEX and G4-FAM on strains ST3 and 116E. Well 3- Amplification plots of G1-HEX, G3-Cy5 and P[4]-TR on strains Wa, P, and DS-1. Well 4- Amplification plots of P[6]-FAM, P[8]-Cy5 and G2-TR on strains ST3, Wa and DS-1. Amplification plots from FAM reporter dye are shown in green, amplification plots from HEX reporter dye are shown in orange, amplification plots from Texas Red reporter dye are shown in red and amplification plots from Cy5 reporter dye are shown in blue.

**Table 2 Rotavirus genotyping on clinical samples using qRT-PCR assay and compared to RT-PCR and sequencing as gold standard.**

| Well 1A | RotaTeq® VP6 gene sequencing | | |
|---|---|---|---|
| **RotaTeq® VP6 HEX qRT-PCR assay** | Positive | Negative | **Total** |
| Positive | 83 | 0 | 83 |
| Negative | 0 | 770 | 770 |
| **Total** | 83 | 770 | 853 |
| **Well 1B** | **Xeno or MS2 RNA spiked before extraction** | | |
| **Xeno or MS2 TR qRT-PCR assays** | Positive | Negative | **Total** |
| Positive | 729 | 0 | 729 |
| Negative | 0 | 124 | 124 |
| **Total** | 729 | 124 | 853 |
| **Well 1C** | **Rotarix® NSP2 gene sequencing** | | |
| **Rotarix® NSP2 Cy5 qRT-PCR assay** | Positive | Negative | **Total** |
| Positive | 39 | 0 | 39 |
| Negative | 0 | 814 | 853 |
| **Total** | 39 | 814 | 853 |
| **Well 1D** | **VP7-G12 gene sequencing** | | |
| **VP7-G12 FAM qRT-PCR assay** | Positive | Negative | **Total** |
| Positive | 173 | 0 | 173 |
| Negative | 2 | 678 | 680 |
| **Total** | 175 | 678 | 853 |
| **Well 2A** | **VP7-G9 gene sequencing** | | |
| **VP7-G9 HEX qRT-PCR assay** | Positive | Negative | **Total** |
| Positive | 109 | 2 | 111 |
| Negative | 1 | 741 | 742 |
| **Total** | 110 | 743 | 853 |
| **Well 2B** | **NSP3-TR RT-PCR** | | |
| **NSP3-TR qRT-PCR assay** | Positive | Negative | **Total** |
| Positive | 833 | 0 | 833 |
| Negative | 0 | 20 | 20 |
| **Total** | 833 | 20 | 853 |
| **Well 2C** | **VP7-G4 gene sequencing** | | |
| **VP7-G4-FAM qRT-PCR assay** | Positive | Negative | **Total** |
| Positive | 79 | 0 | 79 |
| Negative | 0 | 774 | 774 |
| **Total** | 79 | 774 | 853 |

(Continued)

**Table 2** (*continued*).

| Well 3A | VP7-G1 gene sequencing | | |
|---|---|---|---|
| **VP7-G1 HEX qRT-PCR assay** | Positive | Negative | **Total** |
| Positive | 161 | 0 | 161 |
| Negative | 0 | 692 | 692 |
| **Total** | 161 | 692 | 853 |
| **Well 3B** | **VP4-P4 gene sequencing** | | |
| **VP4-P4 TR qRT-PCR assay** | Positive | Negative | **Total** |
| Positive | 103 | 0 | 103 |
| Negative | 0 | 750 | 750 |
| **Total** | 103 | 750 | 853 |
| Well 3C | VP7-G3 gene sequencing | | |
| **VP7-G3 Cy5 qRT-PCR assay** | Positive | Negative | **Total** |
| Positive | 143 | 0 | 143 |
| Negative | 1 | 709 | 710 |
| **Total** | 144 | 709 | 853 |
| **Well 4A** | **VP7-G2 gene sequencing** | | |
| **VP7-G2 TR qRT-PCR assay** | Positive | Negative | **Total** |
| Positive | 110 | 0 | 110 |
| Negative | 0 | 743 | 743 |
| **Total** | 110 | 743 | 853 |
| Well 4B | VP4-P8 gene sequencing | | |
| **VP4-P8 Cy5 qRT-PCR assay** | Positive | Negative | **Total** |
| Positive | 596 | 0 | 596 |
| Negative | 7 | 250 | 257 |
| **Total** | 603 | 250 | 853 |
| **Well 4C** | **VP4-P6 gene sequencing** | | |
| **VP4-P6 FAM qRT-PCR assay** | Positive | Negative | **Total** |
| Positive | 56 | 0 | 56 |
| Negative | 0 | 797 | 797 |
| **Total** | 56 | 797 | 853 |

(RO1845 and CC425). Thus, the RotaTeq® VP6 HEX qRT-PCR exhibited 100% sensitivity and 100% specificity with a PPV of 100% and NPV of 100% (Table 3). Using a ten-fold dilution series of RotaTeq® VP6 dsRNA transcript, RotaTeq® VP6 HEX singleplex qRT-PCR could detect the template in the range of $1.1 \times 10^7$ copies per reaction to 1.1 copies per reaction corresponding to Cq values of 15.1 to 37.9, respectively. The plot of log transcript copy number versus Cq values indicated a linear correlation with a $R^2$ value of 0.9988. The efficiency of RotaTeq® VP6 HEX singleplex qRT-PCR was calculated to be 102% with a

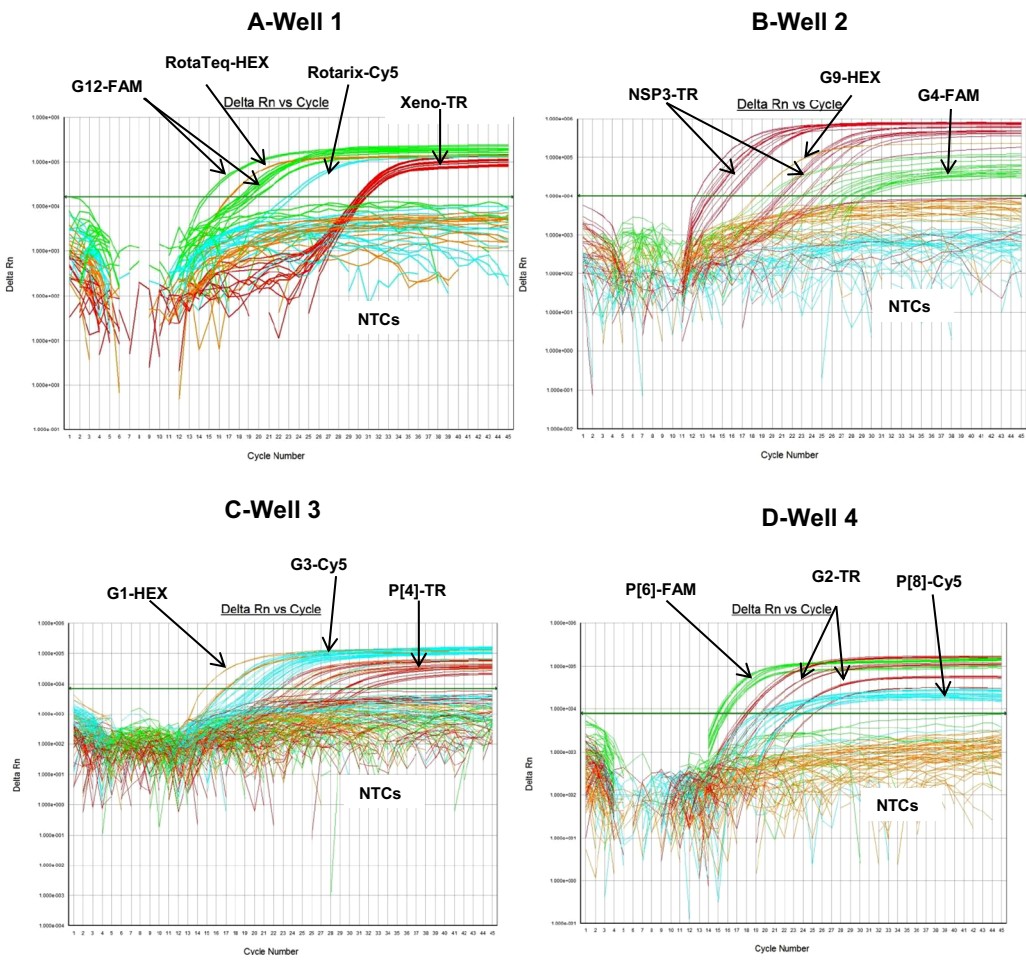

**Figure 2 Performance of 4 well multiplex qRT-PCR assay on sequence confirmed clinical samples.** (A) Well 1- Amplification plots of Xeno-TR, RotaTeq®-HEX, Rotarix®-Cy5, and G12-FAM qRT-PCR on clinical samples spiked with Xeno. (B) Well 2- Amplification plots of NSP3-TR, G9-HEX and G4-FAM on clinical samples. (C) Well 3- Amplification plots of G1-HEX, P[4]-TR and G3-Cy5 assays on clinical samples. (D) Well 4- Amplification plots of G2-TR, P[8]-Cy5 and P[6]-FAM on clinical samples. Amplification plots from FAM reporter dye are shown in green, amplification plots from HEX reporter dye are shown in orange, amplification plots from Texas Red reporter dye are shown in red and amplification plots from Cy5 reporter dye are shown in blue.

limit of detection of 1 copy (Table 3, Figs. S1A and S1B). RotaTeq® VP6 HEX qRT-PCR in Well 1 multiplex reaction could detect the template in the range of $1.1 \times 10^7$ copies per reaction to $1.1 \times 10^1$ copies per reaction with an efficiency of 93% and a limit of detection of 10 copies per reaction (Table 3, Figs. 3A and 3B).

### Xeno or MS2-TR

The Xeno or MS2-TR qRT-PCRs detected Xeno or MS2 RNA respectively in all 729 samples of different genotypes spiked with either Xeno or MS2 RNA. The Xeno or MS2-TR qRT-PCRs showed no amplification in samples that were not spiked with Xeno or MS2 RNA (n = 124) (Table 2-Well 1B). When tested using laboratory
**Table 3 Performance characteristics of rotavirus genotyping qRT-PCR assay.**

| Multiplex qRT-PCR Wells | qRT-PCR assay | Sensitivity (%) | Specificity (%) | PPV (%) | NPV (%) | Limit of detection (copies) | | Efficiency (%) | |
|---|---|---|---|---|---|---|---|---|---|
| | | | | | | Singleplex | Multiplex | Singleplex | Multiplex |
| Well 1 | RotaTeq®-VP6 HEX | 100 | 100 | 100 | 100 | 1 | 10 | 102 | 93 |
| | Xeno or MS2 TR | 100 | 100 | 100 | 100 | – | – | – | – |
| | Rotarix®-NSP2 Cy5 | 100 | 100 | 100 | 100 | 2 | 200 | 91 | 93 |
| | VP7-G12 FAM | 98.8 | 100 | 100 | 99.7 | 60 | 600 | 88 | 81 |
| Well 2 | VP7-G9 HEX | 99 | 99.7 | 98.1 | 99.8 | 60 | 600 | 92 | 92 |
| | NSP3 TR | 100 | 100 | 100 | 100 | 2 | 20 | 93 | 96 |
| | VP7-G4 FAM | 100 | 100 | 100 | 100 | 15 | 150 | 90 | 89 |
| Well 3 | VP7-G1 HEX | 100 | 100 | 100 | 100 | 5 | 500 | 95 | 92 |
| | VP4-P[4] TR | 100 | 100 | 100 | 100 | 12 | 120 | 88 | 82 |
| | VP7-G3 Cy5 | 99.3 | 100 | 100 | 99.8 | 4 | 400 | 86 | 87 |
| Well 4 | VP7-G2 TR | 100 | 100 | 100 | 100 | 4 | 400 | 85 | 92 |
| | VP4-P[8] Cy5 | 98.8 | 100 | 100 | 97.2 | 30 | 300 | 89 | 82 |
| | VP4-P[6] FAM | 100 | 100 | 100 | 100 | 400 | 4000 | 86 | 90 |

reference strains (n = 24) and 2 vaccine strains (Rotarix® and RotaTeq®), the Xeno or MS2-TR qRT-PCRs showed amplification with all the strains spiked with either Xeno or MS2 RNA prior to RNA extraction. Thus, the Xeno or MS2-TR qRT-PCRs exhibited 100% sensitivity and 100% specificity with a PPV of 100% and NPV of 100% (Table 3).

### Rotarix® Cy5

The Rotarix® Cy5 qRT-PCR detected Rotarix® NSP2 gene in all sequence-confirmed Rotarix® positive samples (n = 39) tested (Table 2-Well 1C, Fig. 2A). The Rotarix® Cy5 qRT-PCR showed no amplification with samples of other genotypes, with wild-type G1P[8] (n = 794), or with RVA negative samples (n = 20) (Table 2-Well 1C). When tested using laboratory reference strains (n = 24) and 2 vaccine strains (Rotarix® and RotaTeq®), the Rotarix® Cy5 qRT-PCR showed amplification only with Rotarix® vaccine strain, and no amplification was observed with wild-type G1P[8] or with human or animal strains of non-G1P[8] genotypes. Thus, Rotarix® Cy5 qRT-PCR exhibited 100% sensitivity and 100% specificity with a PPV of 100% and NPV of 100% (Table 3). Using a ten-fold dilution series of Rotarix® NSP2 dsRNA transcript, Rotarix® NSP2 Cy5 singleplex qRT-PCR could detect the template in the range of $2.2 \times 10^7$ copies per reaction to 2.2 copies per reaction corresponding to Cq values of 14.5 to 39.5, respectively. The plot of log transcript copy number versus Cq values indicated a linear correlation with a $R^2$ value of 0.9921. The efficiency of Rotarix® Cy5 singleplex qRT-PCR was calculated to be 91% with a limit of detection of 2 copies (Table 3, Figs. S1C and S1D). Rotarix® NSP2 Cy5 qRT-PCR in Well 1 multiplex reaction could detect the template in the range of $2.2 \times 10^7$ copies per reaction to $2.2 \times 10^2$ copies per reaction with an efficiency of 93% and a limit of detection of 200 copies per reaction (Table 3, Figs. 3C and 3D).

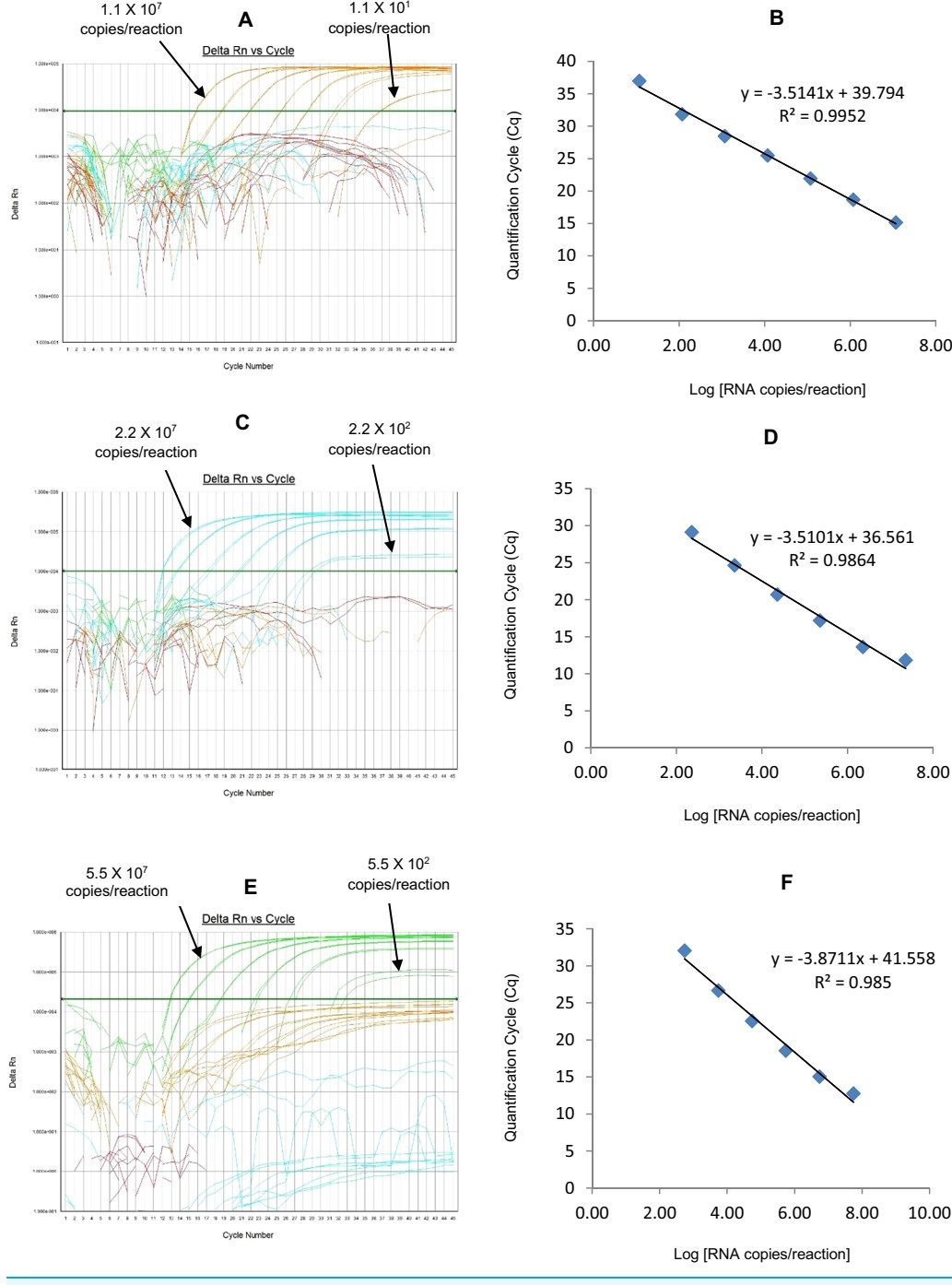

**Figure 3 Efficiency and limit of detection of Well 1 qRT-PCRs in multiplex reaction.** Amplification curves of (A) RotaTeq®-HEX, (C) Rotarix®-Cy5, (E) G12-FAM qRT-PCR using their respective 10-fold serial dilutions of dsRNA transcripts in multiplex reaction and the linear relationship between quantification cycle (Cq) and log transcript copy number per reaction (B) RotaTeq®-HEX, (D) Rotarix®-Cy5, (E) G12-FAM. Graphs showing the Cq value versus the log copy number were fitted with a regression line, and the slope for calculation of efficiency was obtained from the regression line.

### G12 FAM

The G12 FAM qRT-PCR detected G12 genotype in 173 out of 175 sequence-confirmed G12 positive samples tested (Table 2-Well 1D, Fig. 2A). The G12 FAM qRT-PCR showed no amplification with samples of other genotypes (n = 658), or with RVA negative samples (n = 20) (Table 2-Well 1D). When tested using laboratory reference strains (n = 24) and 2 vaccine strains (Rotarix® and RotaTeq®), the G12 FAM qRT-PCR showed amplification only with a strain possessing G12 genotype (L26). Thus, the G12 FAM qRT-PCR exhibited 98.8% sensitivity and 100% specificity with a PPV of 100% and NPV of 99.7% (Table 3). Using a ten-fold dilution series of G12 dsRNA transcript, G12 FAM singleplex qRT-PCR could detect the template in the range of $5.5 \times 10^7$ copies per reaction to $5.5 \times 10^1$ copies per reaction corresponding to Cq values of 15.5 to 37, respectively. The plot of log transcript copy number versus Cq values indicated a linear correlation with a $R^2$ value of 0.9981. The efficiency of G12 FAM singleplex assay was calculated to be 88% with a limit of detection of 60 copies (Table 3, Figs. S1E and S1F). G12 FAM qRT-PCR in Well 1 multiplex reaction could detect the template in the range of $5.5 \times 10^7$ copies per reaction to $5.5 \times 10^2$ copies per reaction with an efficiency of 81% and a limit of detection of 600 copies per reaction (Table 3, Figs. 1E and 1F).

## Well 2 multiplex qRT-PCRs

Multiplex well 2 consisted of G9-HEX, NSP3-TR and G4-FAM qRT-PCR. When tested on lab cultured positive control samples, 116E for G9, ST3 for G4 and both 116E and ST3 for NSP3 genotypes, multiplex well 2 qRT-PCRs showed amplification of all lab strains with NSP3-TR and 116E and ST3 strains with G9 HEX and G4 FAM qRT-PCRs respectively (Fig. 1, Well 2).

### G9 HEX

The G9 HEX qRT-PCR detected G9 genotype in 109 out of 110 sequence-confirmed G9 positive samples tested (Table 2-Well 2A, Fig. 2B). The G9 HEX qRT-PCR showed amplification with 2 samples that were negative by G9 sequencing. The G9 HEX qRT-PCR showed no amplification with samples of other genotypes (n = 721), or with RVA negative samples (n = 20) (Table 2-Well 2A). When tested using laboratory reference strains (n = 24) and 2 vaccine strains (Rotarix® and RotaTeq®), the G9 HEX qRT-PCR showed amplification only with strains possessing G9 genotype (116E, Wi61, US1205, F45 and CC117). Thus, the G9 HEX qRT-PCR exhibited 99% sensitivity and 99.7% specificity with a PPV of 98.1% and NPV of 99.8% (Table 3). Using a ten-fold dilution series of G9 dsRNA transcript, G9 HEX singleplex qRT-PCR could detect the template in the range of $6.0 \times 10^7$ copies per reaction to $6.0 \times 10^1$ copies per reaction corresponding to Cq values of 15.3 to 36.5, respectively. The plot of log transcript copy number versus Cq values indicated a linear correlation with a $R^2$ value of 0.9873. The efficiency of G9 HEX singleplex assay was calculated to be 92% with a limit of detection of 60 copies (Table 3, Figs. S2A and S2B). G9 HEX qRT-PCR in Well 2 multiplex reaction could detect the template in the range of $6.0 \times 10^6$ copies per reaction to $6.0 \times 10^2$ copies per reaction with an efficiency of 92% and a limit of detection of 600 copies per reaction (Table 3, Figs. 4A and 4B).

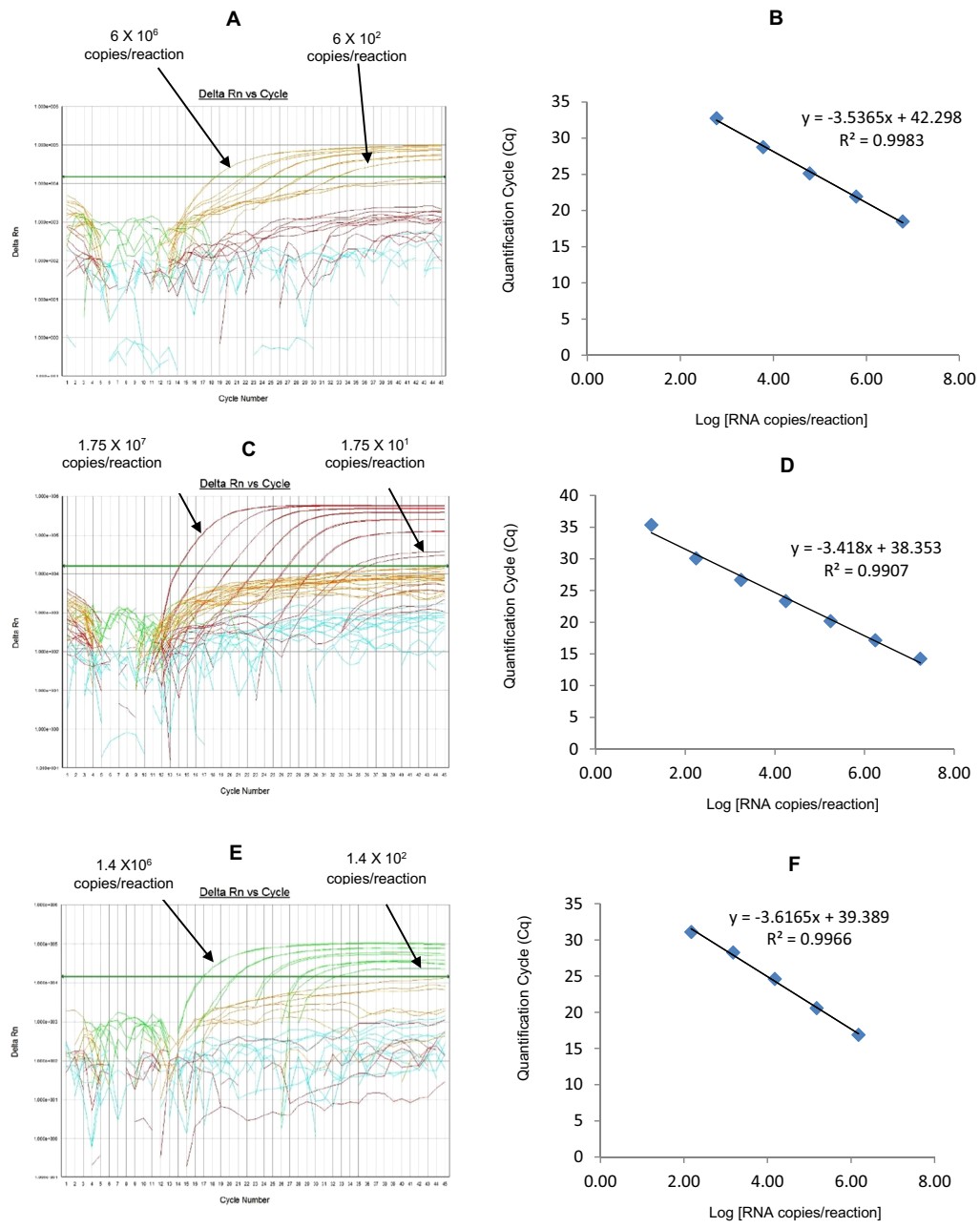

**Figure 4 Efficiency and limit of detection of Well 2 qRT-PCRs in multiplex reaction.** Amplification Curves of (A) G9-HEX, (C) NSP3-TR, (E) G4-FAM using their respective 10-fold serial dilutions of dsRNA transcripts in multiplex reaction and the linear relationship between quantification cycle (Cq) and log transcript copy number per reaction (B) G9-HEX, (D) NSP3-TR, (F) G4-FAM. Graphs showing the Cq value versus the log copy number were fitted with a regression line, and the slope for calculation of efficiency was obtained from the regression line.

### NSP3-TR

The NSP3-TR qRT-PCR detected NSP3 gene in all sequence-confirmed RVA positive samples (n = 833) tested (Table 2-Well 2B, Fig. 2B). The NSP3-TR qRT-PCR showed no amplification with RVA negative samples (n = 20) (Table 2-Well 2B). When tested

using laboratory reference strains (n = 24) and 2 vaccine strains (Rotarix® and RotaTeq®), the NSP3-TR qRT-PCR showed amplification with all strains possessing different genotypes. Thus, the NSP3-TR qRT-PCR exhibited 100% sensitivity and 100% specificity with a PPV of 100% and NPV of 100% (Table 3). Using a ten-fold dilution series of NSP3 dsRNA transcript, NSP3-TR singleplex qRT-PCR could detect the template in the range of $1.75 \times 10^6$ copies per reaction to 1.75 copies per reaction corresponding to Cq values of 12.9 to 34.3, respectively. The plot of log transcript copy number versus Cq values indicated a linear correlation with a $R^2$ value of 0.9881. The efficiency of NSP3-TR singleplex assay was calculated to be 93% with a limit of detection of 2 copies (Table 3, Figs. S2C and S2D). NSP3-TR qRT-PCR in Well 2 multiplex reaction could detect the template in the range of $1.75 \times 10^7$ copies per reaction to $1.75 \times 10^1$ copies per reaction with an efficiency of 96% and a limit of detection of 20 copies per reaction (Table 3, Figs. 4C and 4D).

### G4 FAM

The G4 FAM qRT-PCR detected G4 genotype in all sequence-confirmed G4 positive samples (n = 79) tested (Table 2-Well 2C, Fig. 2B). The G4 FAM qRT-PCR showed no amplification with samples of other genotypes (n = 754), or with RVA negative samples (n = 20) (Table 2-Well 2C). When tested using laboratory reference strains (n = 24) and 2 vaccine strains (Rotarix® and RotaTeq®), the G4 FAM qRT-PCR showed amplification only with strain possessing G4 genotype (ST3 and RotaTeq® vaccine). Thus, the G4 FAM qRT-PCR exhibited 100% sensitivity and 100% specificity with a PPV of 100% and NPV of 100% (Table 3). Using a ten-fold dilution series of G4 dsRNA transcript, G4 FAM singleplex qRT-PCR could detect the template in the range of $1.4 \times 10^7$ copies per reaction to $1.49 \times 10^1$ copies per reaction corresponding to Cq values of 16.9 to 37.5, respectively. The plot of log transcript copy number versus Cq values indicated a linear correlation with a $R^2$ value of 0.9921. The efficiency of G4 FAM singleplex assay was calculated to be 90% with a limit of detection of 15 copies (Table 3, Figs. S2E and S2F). G4 FAM qRT-PCR in Well 2 multiplex reaction could detect the template in the range of $1.4 \times 10^6$ copies per reaction to $1.4 \times 10^2$ copies per reaction with an efficiency of 89% and a limit of detection of 150 copies per reaction (Table 3, Figs. 4E and 4F).

## Well 3 multiplex qRT-PCRs

Multiplex well 3 consisted of G1-HEX, P[4]-TR and G3-Cy5 qRT-PCR. When tested on lab cultured positive control samples, Wa for G1, DS-1 for P[4], P for G3 genotypes, multiplex well 3 qRT-PCR showed amplification with Wa, DS-1 and P strains (Fig. 1, Well 3).

### G1-HEX

The G1-HEX qRT-PCR detected G1 genotype in all sequence-confirmed G1 positive samples (n = 161) tested (Table 2-Well 3A, Fig. 2C). The G1 HEX qRT-PCR showed no amplification with samples of other genotypes (n = 672), or with RVA negative samples (n = 20) (Table 2-Well 3A). When tested using laboratory reference strains (n = 24) and

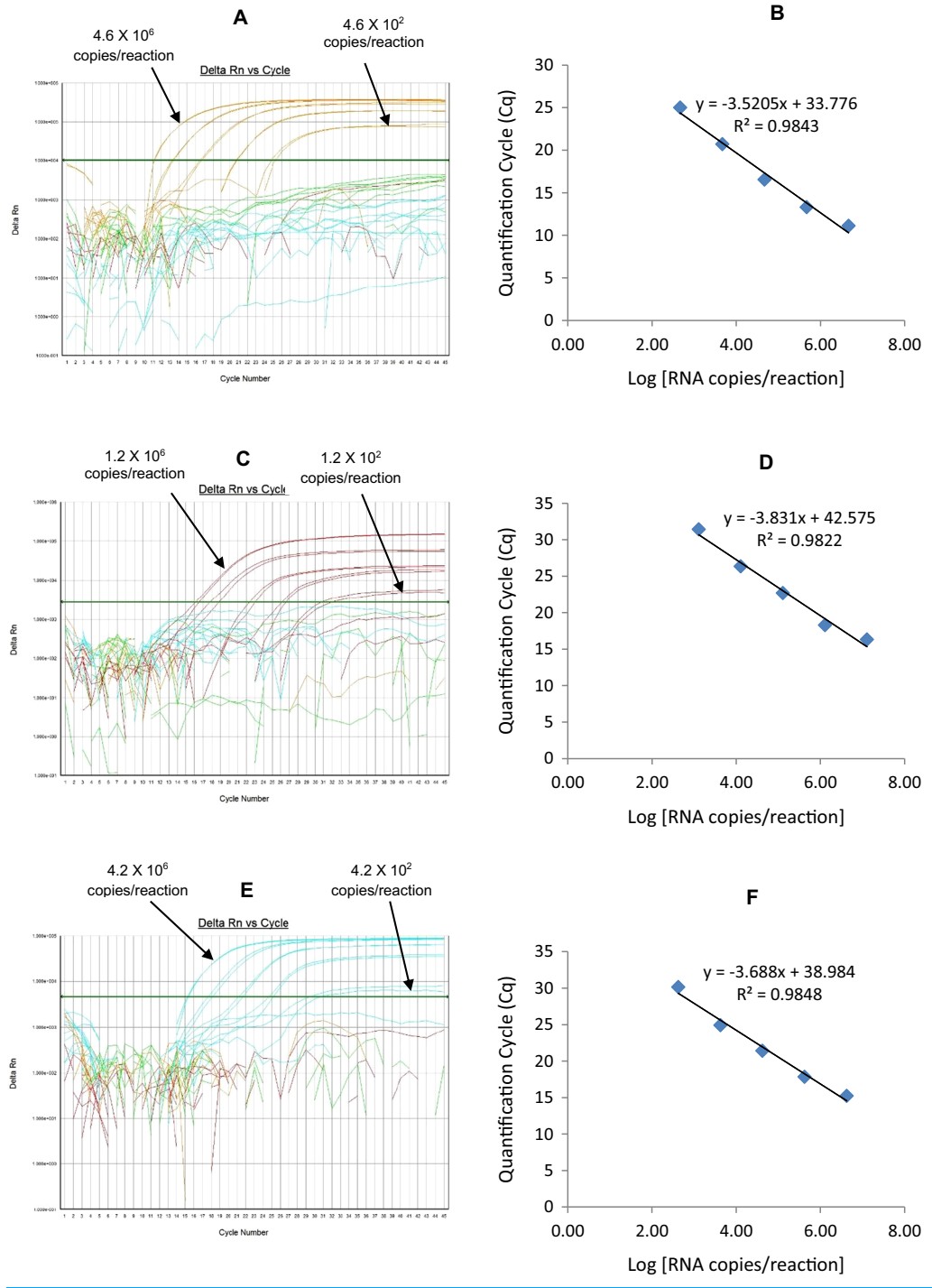

**Figure 5 Efficiency and limit of detection of Well 3 qRT-PCRs in multiplex reaction.** Amplification curves of (A) G1-HEX, (C) P[4]-TR, (E) G3-Cy5 using their respective 10-fold serial dilutions of dsRNA transcripts in multiplex reaction and the linear relationship between quantification cycle (Cq) and log transcript copy number per reaction (B) G1-HEX, (D) P[4]-TR, (F) G3-Cy5. Graphs showing the Cq value versus the log copy number were fitted with a regression line, and the slope for calculation of efficiency was obtained from the regression line.

2 vaccine strains (Rotarix® and RotaTeq®), the G1 qRT-PCR showed amplification only with strains possessing G1 genotype (Wa, RotaTeq® vaccine and Rotarix® vaccine). Thus, the G1 HEX qRT-PCR exhibited 100% sensitivity and 100% specificity with a PPV of 100% and NPV of 100% (Table 3). Using a ten-fold dilution series of G1 dsRNA transcript, G1 HEX singleplex qRT-PCR could detect the template in the range of $4.6 \times 10^6$ copies per reaction to 4.6 copies per reaction corresponding to Cq values of 14 to 35.2, respectively. The plot of log transcript copy number versus Cq values indicated a linear correlation with a $R^2$ value of 0.9933. The efficiency of G1 HEX singleplex assay was calculated to be 95% with a limit of detection of 5 copies (Table 3, Figs. S3A and S3B). G1 HEX qRT-PCR in Well 3 multiplex reaction could detect the template in the range of $4.6 \times 10^6$ copies per reaction to $4.6 \times 10^2$ copies per reaction with an efficiency of 92% and a limit of detection of 500 copies per reaction (Table 3, Figs. 5A and 5B).

### P[4]-TR

The P[4] TR qRT-PCR detected P[4] genotype in all sequence-confirmed P[4] positive samples (n = 103) tested (Table 2-Well 3B, Fig. 2C). The P[4] TR qRT-PCR showed no amplification with samples of other genotypes (n = 730), or with RVA negative samples (n = 20) (Table 2-Well 3B). When tested using laboratory reference strains (n = 24) and 2 vaccine strains (Rotarix® and RotaTeq®), the P[4] qRT-PCR showed amplification only with strains possessing P[4] genotype (DS-1 and L26). Thus, the P[4] TR qRT-PCR exhibited 100% sensitivity and 100% specificity with a PPV of 100% and NPV of 100% (Table 3). Using a ten-fold dilution series of P[4] dsRNA transcript, P[4] TR singleplex qRT-PCR could detect the template in the range of $1.2 \times 10^7$ copies per reaction to $1.2 \times 10^1$ copies per reaction corresponding to Cq values of 17.6 to 39.1, respectively. The plot of log transcript copy number versus Cq values indicated a linear correlation with a $R^2$ value of 0.9977. The efficiency of P[4] TR singleplex assay was calculated to be 88% with a limit of detection of 12 copies (Table 3, Figs. S3C and S3D). P[4] TR qRT-PCR in Well 3 multiplex reaction could detect the template in the range of $1.2 \times 10^6$ copies per reaction to $1.2 \times 10^2$ copies per reaction with an efficiency of 82% and a limit of detection of 120 copies per reaction (Table 3, Figs. 5C and 5D).

### G3-Cy5

The G3 Cy5 qRT-PCR detected G3 genotype in 143 out of 144 sequence-confirmed G3 positive samples tested (Table 2-Well 3C, Fig. 2C). The G3 Cy5 qRT-PCR showed no amplification with samples of other genotypes (n = 689), or with RVA negative samples (n = 20) (Table 2-Well 3C). When tested using laboratory reference strains (n = 24) and 2 vaccine strains (Rotarix® and RotaTeq®), the G3 Cy5 showed amplification with strains possessing human (P, AU-1, RotaTeq® vaccine), human/bovine (CC425) or human/feline reassortant G3 genotype (RO1845). Thus, the G3 Cy5 qRT-PCR exhibited 99.3% sensitivity and 100% specificity with a PPV of 100% and NPV of 99.8% (Table 3). Using a ten-fold dilution series of G3 dsRNA transcript, G3-Cy5 singleplex qRT-PCR could detect the template in the range of $4.2 \times 10^7$ copies per reaction to 4.2 copies per reaction corresponding to Cq values of 14.8 to 36.5, respectively. The plot of log transcript

copy number versus Cq values indicated a linear correlation with a $R^2$ value of 0.9734. The efficiency of G3 Cy5 singleplex assay was calculated to be 86% with a limit of detection of 4 copies (Table 3, Figs. S3E and S3F). G3-Cy5 qRT-PCR in Well 3 multiplex reaction could detect the template in the range of $4.2 \times 10^6$ copies per reaction to $4.2 \times 10^2$ copies per reaction with an efficiency of 87% and a limit of detection of 400 copies per reaction (Table 3, Figs. 5E and 5F).

## Well 4 multiplex qRT-PCR assay

Multiplex well 4 consisted of G2-TR, P[8]-Cy5 and P[6]-FAM qRT-PCR. When tested on lab cultured positive control samples, DS-1 for G2, Wa for P[8] and ST3 for P[6] genotypes, multiplex well 4 qRT-PCR showed amplification with DS-1, Wa and ST3 strains (Fig. 1, Well 4).

### G2-TR

The G2-TR qRT-PCR detected G2 genotype in all sequence-confirmed G2 positive samples (n = 110) tested (Table 2-Well 4A, Fig. 2D). The G2 TR qRT-PCR showed no amplification with samples of other genotypes (n = 723), or with RVA negative samples (n = 20) (Table 2-Well 4A). When tested using laboratory reference strains (n = 24) and 2 vaccine strains (Rotarix® and RotaTeq®), the G2 qRT-PCR showed amplification only with strains possessing G2 genotype (DS-1 and RotaTeq® vaccine). Thus, the G2-TR qRT-PCR exhibited 100% sensitivity and 100% specificity with a PPV of 100% and NPV of 100% (Table 3). Using a ten-fold dilution series of G2 dsRNA transcript, G2 TR singleplex qRT-PCR could detect the template in the range of $3.9 \times 10^6$ copies per reaction to 3.9 copies per reaction corresponding to Cq values of 15.7 to 38.2, respectively. The plot of log transcript copy number versus Cq values indicated a linear correlation with a $R^2$ value of 0.9956. The efficiency of G2-TR singleplex assay was calculated to be 85% with a limit of detection of 4 copies (Table 3, Figs. S4A and S4B). G2 TR qRT-PCR in Well 4 multiplex reaction could detect the template in the range of $3.9 \times 10^7$ copies per reaction to $3.9 \times 10^2$ copies per reaction with an efficiency of 85% and a limit of detection of 400 copies per reaction (Table 3, Figs. 6A and 6B).

### P[8]-Cy5

The P[8] Cy5 qRT-PCR detected P[8] genotype in 596 out of 603 sequence-confirmed P[8] positive samples tested (Table 2-Well 4B, Fig. 2D). The P[8] Cy5 qRT-PCR showed no amplification with samples of other genotypes (n = 230), or with RVA negative samples (n = 20) (Table 2-Well 4B). When tested using laboratory reference strains (n = 24) and 2 vaccine strains (Rotarix® and RotaTeq®), the P[8] assay showed amplification only with strains possessing P[8] genotype (Wa, P, Wi61, F45, CC117, RotaTeq® and Rotarix® vaccines). Thus, the P[8] Cy5 qRT-PCR exhibited 98.8% sensitivity and 100% specificity with a PPV of 100% and NPV of 97.2% (Table 3). Using a ten-fold dilution series of P[8] dsRNA transcript, P[8] Cy5 singleplex qRT-PCR could detect the template in the range of $2.7 \times 10^7$ copies per reaction to $2.7 \times 10^1$ copies per reaction corresponding to Cq values of 13 to 34, respectively. The plot of log transcript copy number versus Cq values indicated a linear correlation with a $R^2$ value

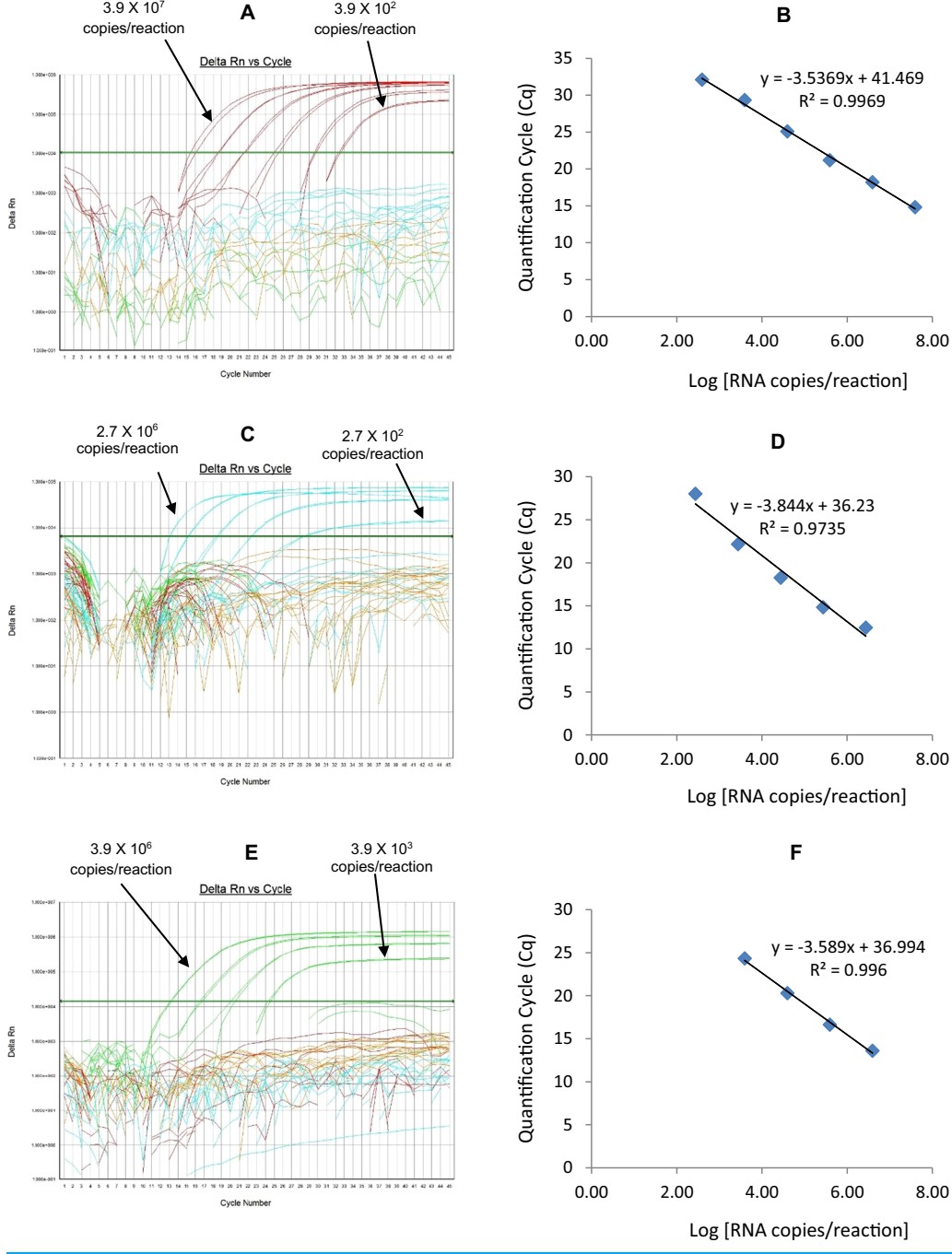

**Figure 6 Efficiency and limit of detection of Well 4 qRT-PCRs in multiplex reaction.** Amplification curves of (A) G2-TR, (C) P[8]-Cy5, (E) P[6]-FAM using their respective 10-fold serial dilutions of dsRNA transcripts in multiplex reaction and the linear relationship between quantification cycle (Cq) and log transcript copy number per reaction (B) G2-TR, (D) P[8]-Cy5, (F) P[6]-FAM. Graphs showing the Cq value versus the log copy number were fitted with a regression line, and the slope for calculation of efficiency was obtained from the regression line.

of 0.9897. The efficiency of P[8] Cy5 singleplex assay was calculated to be 89% with a limit of detection of 30 copies per reaction (Table 3, Figs. S4C and S4D). P[8] Cy5 qRT-PCR in Well 4 multiplex reaction could detect the template in the range of $2.7 \times 10^6$ copies per reaction to $2.7 \times 10^2$ copies per reaction with an efficiency of 82% and a limit of detection of 300 copies per reaction (Table 3, Figs. 6C and 6D).

### P[6] FAM

The P[6] FAM qRT-PCR detected P[6] genotype in all sequence-confirmed P[6] positive samples (n = 56) tested (Table 2-Well 4C, Fig. 2D). The P[6] FAM qRT-PCR assay showed no amplification with samples of other genotypes (n = 777), or with RVA negative samples (n = 20) (Table 2-Well 4C). When tested using laboratory reference strains (n = 24) and 2 vaccine strains (Rotarix® and RotaTeq®), the P[6] FAM showed amplification only with strains possessing P[6] genotype (ST3, US1205 and 1076). Thus, the P[6] FAM qRT-PCR exhibited 100% sensitivity and 100% specificity with a PPV of 100% and NPV of 100% (Table 3). Using a ten-fold dilution series of P[6] dsRNA transcript, P[6] FAM singleplex qRT-PCR could detect the template in the range of $3.9 \times 10^7$ copies per reaction to $3.9 \times 10^2$ copies per reaction corresponding to Cq values of 13.77 to 32.5, respectively. The plot of log transcript copy number versus Cq values indicated a linear correlation with a $R^2$ value of 0.9792. The efficiency of P[6] FAM singleplex assay was calculated to be 86% with a limit of detection of 400 copies (Table 3, Figs. S4E and S4F). P[6] FAM qRT-PCR in Well 4 multiplex reaction could detect the template in the range of $3.9 \times 10^6$ copies per reaction to $3.9 \times 10^3$ copies per reaction with an efficiency of 90% and a limit of detection of 4000 copies per reaction (Table 3, Figs. 6E and 6F).

The RotaTeq® VP6-HEX, Xeno-TR, Rotarix® NSP2-Cy5, NSP3-TR, G4-FAM, G1-HEX, P[4]-TR, G2-TR, and P[6]-FAM qRT-PCRs displayed 100% sensitivity and specificity. G12-FAM qRT-PCR missed two samples that were genotyped as G12 by sequencing (Table 2-Well 1-D), G9 HEX qRT-PCR missed one sample that was genotyped as G9 by sequencing (Table 2-Well 2-A), G3 Cy5 qRT-PCR missed one sample that was genotyped as G3 by sequencing (Table 2-Well 3-C) and P[8] Cy5 qRT-PCR missed seven samples that were genotyped as P[8] by sequencing (Table 2-Well 4-B) probably due to poor sample quality or due to degradation of VP7 and VP4 gene segments. The qRT-PCRs in multiplex reaction showed one log (RotaTeq® VP6-HEX, G12-FAM, NSP3-TR and G4-FAM) or two logs (Rotarix® NSP2-Cy5, G1-HEX, P[4]-TR, G3-Cy5, G2-TR and P[8]-Cy5) difference in the limit of detection as compared to their respective singleplex qRT-PCRs. G9-HEX and P[6]-FAM qRT-PCRs exhibited same limit of detection in both singleplex and multiplex reactions (Table 3). The qRT-PCRs used in the multiplex assay exhibited sensitivity in the range of 98.8% to 100%, specificity in the range of 99.7% to 100%, limit of detection in the range of 10–4000 copies and efficiency in the range of 81% to 96%.

## DISCUSSION

In this study, we have developed and validated a one-step multiplex qRT-PCR assay for detecting and genotyping rotavirus wild-type and vaccine strains (Rotarix® and

RotaTeq®) in stool samples. We propose that the one-step multiplex qRT-PCR assay should be used for rapid screening of samples for detecting rotavirus positive samples, to identify VP7 and VP4 genotype of positive samples and to detect Rotarix® and RotaTeq® vaccine strains or their components in stool samples. A panel of samples could be sequenced for VP7 and VP4 genes by Sanger sequencing to confirm the genotypes obtained by the multiplex qRT-PCR assay. Sequencing of all eleven genes (by Next Gen) could be performed to obtain full genomic characterization of rotavirus positive samples.

In the one-step multiplex qRT-PCR assay, dsRNA denaturation, reverse transcription (RT) and amplification (PCR) were carried in a single tube using r*Tth* with uninterrupted thermal cycling to reduce manipulation of samples, decrease possibility of sample cross-contamination and rapid generation of results. This is the first rotavirus genotyping qRT-PCR assay to use artificial quantitative standards (dsRNA transcripts) to calculate the efficiency and limit of detection of qRT-PCRs. RT-PCR assays are prone to false-negative results due to presence of inhibitors in the stool samples. We have included an internal process control (Xeno or MS2) qRT-PCR in the multiplex screening assay to monitor the efficiency of RNA extracted and to check for PCR inhibitors present in the stool samples. The amplification curves of Xeno or MS2 qRT-PCRs will shift from their normal range of Cq value 28–31 to Cq value >31 in the presence of PCR inhibitors in stool samples or will test negative.

For routine RVA surveillance, we recommend that RNA from stool samples should be spiked with either Xeno or MS2 RNA and extracted using the MagMax 96 Viral RNA Isolation kit (Applied Biosystems, Inc., Foster City, CA, USA) on an automated KingFisher extraction system (Thermo Scientific, Waltham, MA, USA) and screened using the one-step multiplex qRT-PCR assay. Samples that fail to genotype should be re-extracted using MagNA Pure compact RNA extraction kit on MagNA Pure compact instrument (Roche Applied Science, Indianapolis, IN) and screened again using the one step multiplex qRT-PCR assay. Upon comparison of different kits used for RNA extraction from stools, the RNA extracted by MagNA Pure compact instrument yields best quality RNA with least amount of PCR inhibitors (*Esona et al., 2013*). The samples testing positive by Xeno/MS2 and NSP3 qRT-PCRs but negative by other qRT-PCRs should be re-tested by qRT-PCRs in singleplex format. Singleplex qRT-PCRs are more sensitive, due to less competition for reagents and low background florescence than the qRT-PCRs in multiplex format, thus weak positive samples with Cq values >35 cycles could be missed by the multiplex qRT-PCR assay. Samples testing positive by Xeno/ MS2 qRT-PCRs with inhibition (shift in Cq value of Xeno/MS2 amplification curve) but negative by multiplex and singleplex qRT-PCR should be re-tested by singleplex qRT-PCRs using 1:10 and 1:100 dilutions of RNA to dilute out the inhibitors. Sample testing negative by all the above mentioned steps should be reported as non-typeable by the qRT-PCR assay. We recommend that the genotypes of RVA positive samples and all the samples found to contain vaccine strain genes by the screening qRT-PCR assay should be confirmed by Sanger or Next Gen sequencing.

Limitations of this study are that the genotypes of qRT-PCR positive samples with high Cq values (35–40) will be difficult to sequence confirm. Samples with highly divergent G and P genotype strains will be missed by their respective qRT-PCRs. The qRT-PCRs designed in this study included only 6 predominant G-genotypes (G1–4, G9 and G12), 3 predominant P genotypes (P[4], P[6], and P[8]) and one gene specific for each vaccine strain (NSP2 gene for Rotarix® and VP6 gene for RotaTeq®). Thus, this assay won't genotype the samples of G and P genotype combinations not covered in this assay and won't detect reassortants produced by exchange of other segments between wild-type RVA and vaccine strains as described previously (*Rose et al., 2013*; *Bucardo et al., 2012*).

Validation of the one step multiplex qRT-PCR assay on a large panel of sequence confirmed clinical samples, lab cultured strains of various genotype combinations and vaccine strains (RotaTeq® and Rotarix®) has consistently demonstrated that this assay can detect wild-type and vaccine strain (RotaTeq® and Rotarix®) rotavirus in 100% of samples tested. Changing the reporter dye on NSP3 qRT-PCR probe to Texas red, RotaTeq VP6 qRT-PCR probe to HEX and Rotarix NSP2 probe to Cy5 in place of FAM as used in published NSP3, RotaTeq VP6 and Rotarix assays, didn't have a significant effect on the performance of these assays in singleplex reactions. Also, this assay can correctly genotype 99.0% to 99.5% of rotavirus positive samples for three common P (VP4) genotypes (P[4], P[6] and P[8]) and for six common G (VP7) genotypes (G1-G4, G9 and G12) respectively.

## CONCLUSION

This is the first one-step multiplex qRT-PCR assay developed for detecting and genotyping wild-type rotavirus and vaccine strains (Rotarix® and RotaTeq®) in stool samples. The assay developed and validated in this study can be used for rapid detection of rotavirus, characterization of VP7 and VP4 genotypes and detection of Rotarix® and RotaTeq® vaccine strains along with an internal process control. This assay will be useful for high-throughput screening of stool samples in surveillance studies for monitoring the rotavirus strain prevalence, determining the frequency of vaccine strains and vaccine derived strains associated with AGE.

## ACKNOWLEDGEMENTS

We would like to thank Ms. Leanne Ward for her critical review of the manuscript. We would also like to thank Ms. Alice Williams and Mr. Young Ye for providing technical help. We also would like to thank Ms. Jennifer Hull for providing rotavirus laboratory strains used as positive controls in this study.

### Funding

Funding was provided by CDC core activity funds. The funders had no role in study design, data collection and analysis, decision to publish, or preparation of the manuscript.

## Competing Interests

The authors declare that they have no competing interests.

## Author Contributions

- Rashi Gautam conceived and designed the experiments, performed the experiments, analyzed the data, contributed reagents/materials/analysis tools, wrote the paper, prepared figures and/or tables, reviewed drafts of the paper.
- Slavica Mijatovic-Rustempasic performed the experiments, analyzed the data, contributed reagents/materials/analysis tools, wrote the paper, prepared figures and/or tables, reviewed drafts of the paper.
- Mathew D Esona performed the experiments, analyzed the data, contributed reagents/materials/analysis tools, wrote the paper, prepared figures and/or tables, reviewed drafts of the paper.
- Ka Ian Tam performed the experiments, analyzed the data, contributed reagents/materials/analysis tools, wrote the paper, prepared figures and/or tables, reviewed drafts of the paper.
- Osbourne Quaye performed the experiments, analyzed the data, contributed reagents/materials/analysis tools, wrote the paper, prepared figures and/or tables, reviewed drafts of the paper.
- Michael D Bowen conceived and designed the experiments, analyzed the data, contributed reagents/materials/analysis tools, wrote the paper, prepared figures and/or tables, reviewed drafts of the paper.

## Human Ethics

The following information was supplied relating to ethical approvals (i.e., approving body and any reference numbers):

This work was approved by the CDC as a public health non-research study.

## Data Deposition

The data files have been uploaded as Supplementary Information.

## Supplemental Information

Supplemental information for this article can be found online at http://dx.doi.org/10.7717/peerj.1560#supplemental-information.

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
