# Peer review of "One-step multiplex real-time RT-PCR assay for detecting and genotyping wild-type group A rotavirus strains and vaccine strains (Rotarix® and RotaTeq®) in stool samples"

_PeerJ, doi:10.7717/peerj.1560_

## Round 0.1 · original submission · Minor Revisions

· Academic Editor

Minor Revisions

Dear Author,

Thank you for submitting your manuscript to Peer J. As you can see, two expert reviewers have provided comments that need to be addressed. Please make those changes and provide the response to the reviewers comments.

Reviewer 1 ·

Basic reporting

This is a very interesting paper about genotyping of Rotavirus by quantitative RT-PCR.

The authors go to great lengths to optimize and validate their 4-well qPCR, and the article merits to be published. Nevertheless, I have some concerns regarding data analysis (figures 3-6 and supp. figures). The authors should re-evaluate their results before final acceptance of the paper.

Major comments:

Figures 3-6 and Supp. Fig. 1-4: Please do not indicate fold-dilutions (Fig. A, C and E); indicate copies/reaction instead.

In Fig. 3-6 and Supp. Fig. 1-4 A, C, E: it would help the reader if only the upper part of the figures is depicted and expanded (e.g. range of deltaRn only 4-5 logs).

Lowest dilution (highest concentration). Valid for all figures: The end of the baseline seems to be set too far on the right side. This leads to overestimation of the Cq-values, overestimation of the efficiency and underestimation of the correlation coefficient. How were the baselines set? Automatic (optional on the ABI7500Fast) of did the authors set a fix baseline? This seems to be a problem in Supp. Fig 3A and 3E

Threshold in Fig. 3-6: is set correctly only for figure 6A and 6E. For all other figures, the threshold is set too high. This leads to Cq overestimation of the highest dilution (lowest concentration), and consequently to underestimation of the efficiency and underestimation of the correlation coefficient. The threshold should not be set so that the Cq-value defines a point of the curve that is already out of the log-linear amplification.

Figure 3C; 4A; Supp Fig. 1E; 4C The last dilution does not seem positive to me, rather an unspecific amplification? The authors should correct the sensitivity here (10-fold lower)

Figure 4E; 5A; 6E; Supp Fig. 2E;3C: the replicates of the last dilution are bad. I do not trust this result.

In general, I’d rather state that the sensitivity of the multiplex reactions is 10-fold lower than reported here.


Minor comments:

Overall: Please write Cq (Quantification cycle) instead of Ct.

Line 139: multiple alignments: which software was used? Constraints?

Line 165: 10% suspension: weight/volume (stool)?

Line 166: 10% suspension of reference strain: please clarify

Line 167: 108 copies/µl?

Line 168: 108 units/µl?

Line 198: »singleplex» not «singeplex»

Lines 244-251:
«Each sample was tested by multiplex qRT-PCR assay in four separate reactions, the reaction mixture composition (25 µL/well) for each well was unique, with well 4 using twice the concentration of rTth enzyme than wells 1, 2 and 3. The master mix for each well contained 5 µLof 5X EZ buffer, 2.4 µM dNTPs, 2.5 mM Mn(OC)2, 2.5 U/µLrTth polymerase (wells 1, 2 and 3) or 5.0 U/µLrTth polymerase (Well 4), forward primer, reverse primer and probe at final concentrations specific for each component assay (Table 1). Nuclease free water was added to each well to make final volume of mastermix/well to 23 µL. Two microliters of extracted RNA were added to each sample well and 2 µLof nuclease free water was added to no template control (NTC) wells»
I suggest the authors change the sentence to «Each sample was tested by multiplex qRT-PCR assay in four separate reactions. The master mix for each well contained 5 µLof 5X EZ buffer, 2.4 µM dNTPs, 2.5 mM Mn(OC)2, 2.5 U/µLrTth polymerase (wells 1, 2 and 3) or 5.0 U/µLrTth polymerase (Well 4), forward primer, reverse primer and probe at final concentrations specific for each component assay (Table 1); and 2ul template in a 25ul final volume»

Lines 262-266 «. Initially the allocation of qRT-PCRs in wells 1-4 was random, but later some of the qRT-PCRs were switched between wells in order to maximize amplification of all the qRT-PCRs with least background and minimize cross reactivity. After selection of reporter dye and designation of a well number to each qRT-PCR; …»
I suggest the authors change the sentence to «After selection of the system combinations with a minimal cross-reactivity and lowest background, 13 qRT-PCRs were validated …»

Line 271 which standard procedures? Please clarify

Line 301 and following «10-3 dilution (1.1 X 107 copies per reaction) to 10-10 dilution (1.1 copies per reaction)» please indicate only copy numbers.

Line 516 «qRT-PCR missed two samples that were genotyped as G12 by Next Gen sequencing» This was not mentioned in the material and methods section? Please clarify.

Experimental design

No comments

Validity of the findings

No comments

Reviewer 2 ·

Basic reporting

No Comments

Experimental design

No Comments

Validity of the findings

No Comments

Additional comments

The reviewer wish to congratulate the authors for their work.

The article is well written and the objectives have been achieved following proper methodology, samples and controls.

The detection and genotyping of rotavirus is of great interest and new detection and genotyping methodologies are welcome by the scientific community. The authors have managed to develop a 4 reactions one-step multiplexed qRT-PCR that is able to detect the two main rotavirus vaccines (rotarix and rotateq), the three more predominant P types (P[4], P[6] and P[8]), the 6 more predominant G types (G1 to G4, G9 and G12) and also included a rotavirus detection control previously published and an internal process control (IPC). On top of including the appropriate controls the authors have validated each primer/probe set both in singleplex and in multiplex qPCR. The authors have also utilized dsRNA as template to obtain the efficiency of each of the probes. For the validation of the assay the authors have utilized 853 clinical samples (sequenced confirmed), 24 lab cultured strains and 2 vaccines.

The only disadvantage that can be found is that the setting up of the 4 reactions and the interpretation of the results can be confusing for new users of the technology due to the complexity of the assay.

---

## Round 0.2 · accepted · Accept

· Academic Editor

Accept

Dear Rashi Gautam,

I have reviewed your revision, and I am pleased to inform you that your manuscript "One-step multiplex real-time RT-PCR assay for detecting and genotyping wild-type group A rotavirus strains and vaccine strains (Rotarix® and RotaTeq® ) in stool samples" (#2015:09:6865:1:1:REVIEW), has been accepted for publication.